# Antibody Conjugates Bispecific for Pollen Allergens and ICAM-1 with Potential to Prevent Epithelial Allergen Transmigration and Rhinovirus Infection

**DOI:** 10.3390/ijms24032725

**Published:** 2023-02-01

**Authors:** Christina Weichwald, Ines Zettl, Isabella Ellinger, Katarzyna Niespodziana, Eva E. Waltl, Sergio Villazala-Merino, Daniel Ivanov, Julia Eckl-Dorna, Verena Niederberger-Leppin, Rudolf Valenta, Sabine Flicker

**Affiliations:** 1Division of Immunopathology, Institute for Pathophysiology and Allergy Research, Center for Pathophysiology, Infectiology and Immunology, Medical University of Vienna, 1090 Vienna, Austria; 2Division of Cellular and Molecular Pathophysiology, Institute for Pathophysiology and Allergy Research, Center for Pathophysiology, Infectiology and Immunology, Medical University of Vienna, 1090 Vienna, Austria; 3Department of Otorhinolaryngology, Medical University of Vienna, 1090 Vienna, Austria; 4Karl Landsteiner University of Health Sciences, 3500 Krems, Austria; 5National Research Centre (NRC) Institute of Immunology Federal Medical-Biological Agency (FMBA) of Russia, 115478 Moscow, Russia; 6Laboratory for Immunopathology, Department of Clinical Immunology and Allergy, Sechenov First Moscow State Medical University, 119435 Moscow, Russia

**Keywords:** allergy, allergen, rhinovirus, antibody, ICAM-1, respiratory epithelium, topical treatment

## Abstract

Allergy and rhinovirus (RV) infections are major triggers for rhinitis and asthma, causing a socioeconomic burden. As RVs and allergens may act synergistically to promote airway inflammation, simultaneous treatment strategies for both causative agents would be innovative. We have previously identified the transmembrane glycoprotein intercellular adhesion molecule 1 (ICAM-1) as an anchor for antibody conjugates bispecific for ICAM-1 and *Phleum pratense* (Phl p) 2, a major grass pollen allergen, to block allergen transmigration through the epithelial barrier. Since ICAM-1 is a receptor for the major group RVs, we speculated that our bispecific antibody conjugates may protect against RV infection. Therefore, we created antibody conjugates bispecific for ICAM-1 and the major grass pollen allergen Phl p 5 and analyzed their capacity to affect allergen penetration and RV infection. Bispecific antibody conjugates significantly reduced the trans-epithelial migration of Phl p 5 and thus the basolateral Phl p 5 concentration and allergenic activity as determined by humanized rat basophilic leukemia cells and inhibited RV infection of cultured epithelial cells. A reduction in allergenic activity was obtained only through the prevention of allergen transmigration because the Phl p 5-specific IgG antibody did not block the allergen–IgE interaction. Our results indicate the potential of allergen/ICAM-1-specific antibody conjugates as a topical treatment strategy for allergy and RV infections.

## 1. Introduction

Almost 30% of the world population is affected by an Immunoglobulin E (IgE)-associated allergy, with *Poaceae* pollen as one of the leading aeroallergen sources worldwide [1,2]. Allergic manifestation elicited by airborne allergens includes allergic rhinitis and asthma. Upper and lower airway inflammation caused by allergy and virus infections, in particular by rhinovirus (RV), is a global health problem [3]. Affected individuals are often exposed to airborne allergens and respiratory viruses at the same time. Since the respiratory mucosa is the main entry site for both environmental triggers (i.e., allergens and viruses), it plays an important role in host defense and regulation of inflammation in airway tissues. It is well-documented that allergic sensitization, allergen exposure and RV infections are mutually supportive in inducing rhinitis and asthma [4,5,6,7]. Allergic airway inflammation impairs immune responses against RVs, rendering allergic subjects more sensitive to RV infections and RV-triggered asthma [7]. One of the causes explaining the defective antiviral response during allergic inflammation is the upregulated expression of intercellular adhesion molecule 1 (ICAM-1), the receptor for major group RVs, leading to an increased susceptibility to RV infection [8,9]. On the other hand, RV infections damage the barrier function of the respiratory epithelial cell layer and facilitate trans-epithelial allergen penetration, thereby increasing submucosal allergen concentrations [10,11,12]. Consequently, local and systemic allergen-specific IgE levels are boosted, which may potentially aggravate allergic inflammation [13,14,15,16].

Several strategies including allergen-specific immunotherapy (AIT) are available to prevent allergic rhinitis and also the progression to allergic asthma, one of the most severe manifestations of respiratory allergy [1]. However, no vaccines for either prophylactic or acute treatment of RV infection have been approved so far [17]. The evidence that allergic inflammation and RV infection act synergistically in triggering airway inflammation initiated efforts to develop successful treatments that may be effective for allergic as well as RV-induced rhinitis and asthma [18]. A combined vaccine based on the RV-derived surface protein viral protein (VP)1 and a pollen allergen peptide was found to induce antibodies that inhibited allergen-induced basophil degranulation and protected cultured human epithelial cells against RV infection [19]. Furthermore, T helper (T_H)_2-targeting biologics were shown to be beneficial for reducing harmful T_H_2-related inflammation and for restoring a proper interferon response as well as increasing antiviral immunity [6]. Previously, we have developed bispecific antibody conjugates to capture allergens on the mucosal surface and block the transmigration of allergens through the epithelial barrier [20]. Our antibody conjugates consisted of allergen-specific blocking human antibodies and antibodies specific for ICAM-1 [20]. ICAM-1 was selected as an anchor for these bispecific antibody conjugates because it is highly expressed on epithelial cells of allergic patients, especially under inflammatory conditions [21]. It was demonstrated that bispecific antibody conjugates remained bound to ICAM-1 on the apical side of airway epithelial cells in a sustained manner. We speculated that these bispecific antibody conjugates may also protect against RV infections because it has been shown that the administration of a human ICAM-1-specific antibody in a murine RV-asthma model inhibited the entry of RV into epithelial cells, and thus inhibited exacerbations of RV-induced lung inflammation [22]. To prove our hypothesis, we formed bispecific antibodies for the major grass pollen allergen Phl p 5 and ICAM-1 through chemical conjugation as previously described by [20], and investigated, for the first time, if these conjugates prevent RV infection in vitro. Furthermore, we addressed the question whether bispecific antibody conjugates can reduce allergic inflammation by preventing trans-epithelial allergen migration without blocking the allergen–IgE interaction. Our results revealed that a bispecific antibody conjugate based on an allergen-specific IgG antibody that does not interfere with the allergen–IgE binding inhibited trans-epithelial allergen migration, basophil activation and RV infections in vitro. Our findings indicate the potential of topical application of allergen/ICAM-1-specific antibody conjugates for the treatment of allergen- and RV-induced airway inflammation.

## 2. Results

### 2.1. αPhl p 5/αICAM-1 Antibody Conjugates Specifically Bound to Phl p 5 and Human ICAM-1

Antibody conjugates (αPhl p 5/αICAM-1) formed via streptavidin–biotin coupling of human αPhl p 5-Immunoglobulin G (IgG)*Strep and mouse αICAM-1-IgG*Bio were tested if they retained their specificity for both antigens, i.e., Phl p 5 and human ICAM-1 (Figure 1). Binding of the bispecific αPhl p 5/αICAM-1 conjugates to Phl p 5 (Figure 1A) and human ICAM-1 (Figure 1B) was demonstrated by detecting either the human αPhl p 5-IgG*Strep (with anti-human Fragment antigen-binding (F(ab’)_2_), green bars) or the mouse αICAM-1-IgG*Bio (with anti-mouse IgG, blue bars) (Figure 1C,D). Both chosen detection systems confirmed specific recognition of Phl p 5 and ICAM-1 in a dose-dependent manner (i.e., 5 µg/mL, 1 µg/mL and 0.2 µg/mL) by αPhl p 5/αICAM-1. Bispecific conjugates showed specific reactivity to both antigens compared to the binding of αPhl p 5-IgG*Strep and αICAM-1-IgG*Bio for each respective antigen, indicating that both antibodies were still fully functional when conjugated. No recognition of Phl p 5 was found with either αPhl p 5-IgG*Strep detected by anti-mouse IgG or αICAM-1-IgG*Bio independent of the applied detection system (Figure 1A). Similarly, no binding to ICAM-1 was observed with αICAM-1-IgG*Bio when using anti-human F(ab’)_2_ or with αPhl p 5-IgG*Strep applying any of the detection systems (Figure 1B). No signal was found when Phl p 5 and ICAM-1 were incubated with detection antibodies alone (Figure 1A,B: buffer).

### 2.2. Phl p 5 Was Immobilized on the Surface of Bronchial Epithelial Cells via αPhl p 5/αICAM-1 Antibody Conjugates

After having proven the retained specificity of the generated antibody conjugate, we next tested whether the bispecific conjugate recognized ICAM-1 expressed on the surface of human epithelial cells. Therefore, we used the bronchial epithelial cell line 16HBE14o-, a well-established surrogate model for investigations of barrier function and trans-epithelial migration. We first verified the expression of human ICAM-1 on the surface of the 16HBE14o- cells through flow cytometry using αICAM-1-IgG*Bio (Figure 2A, upper panel, white histogram peak). Positive staining by αICAM-1-IgG*Bio was observed for 98.5% of the live cell population (Figure 2A, lower panel). No staining was detected with the biotinylated isotype control antibody (Figure 2A, upper panel, gray histogram peak).

To examine the immobilization of Phl p 5 on the cell surface, 16HBE14o- cells were pre-incubated with αPhl p 5/αICAM-1 conjugates and, after washing, exposed to different amounts of Phl p 5 (1 µg/50 µL, 2 µg/50 µL and 5 µg/50 µL). Using Phl p 5-specific rabbit antibodies and Alexa Fluor 405-labeled goat anti-rabbit antibodies, successively, immobilization of Phl p 5 onto the cell surface via αPhl p 5/αICAM-1 was demonstrated (Figure 2B, upper panel, white peaks). Scatter plot analysis showed that the immobilization level of Phl p 5 was the lowest when cells were exposed to 1 µg Phl p 5 (around 31% of the cells had Phl p 5 bound to their cell surface) and reached its peak (47%) when the amount of Phl p 5 was increased to 2 µg. This level could not be further enhanced when 5 µg Phl p 5 was applied, indicating a saturation of binding sites of αPhl p 5/αICAM-1 conjugates (Figure 2B, lower panel). A small percentage of the cells also exhibited a positive signal with the isotype control, which seemed to be derived from unspecific binding by rabbit pre-immune serum to the surface of 16HBE14o- to a minor extent (Figure 2B, upper panel, gray peaks).

Phl p 5 binding to αPhl p 5/αICAM-1 conjugates on epithelial cells was visualized by wide-field immunofluorescence microscopy. The images showed co-localization (Figure 3A, yellow, merge) of αPhl p 5/αICAM-1 (Figure 3A, green, Alexa Fluor 488) and Phl p 5 (Figure 3A, red, Alexa Fluor 568) on the cell surface of 16HBE14o- cells after incubation of cells with preformed αPhl p 5/αICAM-1 + Phl p 5 complexes for 1 h at 4 °C. When Phl p 5 (Figure 3B) was omitted, no binding of Alexa Fluor 568 anti-rabbit antibodies was observed. When αPhl p 5/αICAM-1 conjugates were not applied to the cells, neither αPhl p 5/αICAM-1 nor Phl p 5 could be visualized (Figure 3C). To investigate a possible internalization of the αPhl p 5/αICAM-1 + Phl p 5 complex over a given time course and to come as close as possible to physiological conditions, we also analyzed the cells incubated with preformed complexes for 24 h at 37 °C through confocal microscopy (Appendix A). Saponin (0.05%) was used as a detergent in all experiments to enable binding of fluorescently labeled detection antibodies to any αPhl p 5/αICAM-1 or Phl p 5 molecules localized in intracellular compartments. However, allergen and antibody conjugates were only detected at the apical plasma membrane of the epithelial cells (Appendix A), demonstrating a lack of αPhl p 5/αICAM-1 and Phl p 5 uptake into epithelial cells within 24 h.

### 2.3. αPhl p 5/αICAM-1 Conjugates Prevented Transmigration of Phl p 5 through Bronchial Epithelial Cell Layers, Leading to Decreased Basophil Activation

As we observed an immobilization of Phl p 5 by αPhl p 5/αICAM-1 on the cell surface, we next explored if αPhl p 5/αICAM-1 conjugates were able to prevent the apical-to-basolateral migration of Phl p 5 through a layer of bronchial epithelial cells. The 16HBE14o- cells grown on Transwell permeable supports were loaded with or without αPhl p 5/αICAM-1 conjugates in the apical medium. After removing unbound αPhl p 5/αICAM-1 by washing, Phl p 5 was added to the apical medium, and its penetration of the layer into the basolateral medium was measured (Figure 4A). The apical addition of αPhl p 5/αICAM-1 significantly reduced the migration of Phl p 5 into the basolateral medium over the analyzed period (i.e., 24 h) (Figure 4A, green bars, 24 h **+**) when compared to Phl p 5 quantities measured in basolateral media without the addition of αPhl p 5/αICAM-1 to the cell layer (Figure 4A, green bars, 24 h **−**). The trans-epithelial migration of Phl p 2, an unrelated grass pollen allergen, was not inhibited by αPhl p 5/αICAM-1 (Figure 4B, green bars, 24 h +/−), indicating that the measured effect of the αPhl p 5/αICAM-1 conjugate is allergen-specific.

In parallel, by studying the availability of Phl p 5 bound to αPhl p 5/αICAM-1 conjugates in basolateral media, it was revealed that no relevant amounts of αPhl p 5/αICAM-1 + Phl p 5 complexes were detected in the apical and basolateral media (Figure 4C, green bars, 24h +/−). Preformed αPhl p 5/αICAM-1 + Phl p 5 complexes served as a positive control, confirming the functionality of the established assay.

To test the effect of reduced allergen quantities in the basolateral media by cell-bound αPhl p 5/αICAM-1 on the degree of effector cell activation, basophil activation assays were performed. Rat basophilic leukemia cells expressing the human FcεRI were loaded with IgE from patients sensitized to Phl p 5. Basolateral media from 16HBE14o- cell layers that had been coated with αPhl p 5/αICAM-1 conjugates (Figure 5, **+** αPhl p 5/αICAM-1) elicited a significantly reduced release of ß-hexosaminidase as compared to basolateral media from 16HBE14o- cells without coating with αPhl p 5/αICAM-1 (Figure 5, **− **αPhl p 5/αICAM-1).

### 2.4. The Ability to Inhibit Basolateral Basophil Activation Did Not Depend on the Capacity of Allergen-Specific IgG Antibodies to Block Allergen–IgE Interaction

As already reported earlier, we wanted to confirm that the αPhl p 5-IgG*Strep antibody as one part of the here-described bispecific αPhl p 5/αICAM-1 conjugate was a human IgG antibody that does not block patients’ IgE binding to Phl p 5 [23]. Therefore, we conducted inhibition Enzyme-linked immunosorbent assay (ELISA) experiments and compared the inhibitory potential of the Phl p 5-specific IgG_1_ antibody (Phl p 5 IgG_1_) to the protective capacity of a Phl p 2-specific IgG_1_ antibody (Phl p 2 IgG_1_), to Phl p 5 and Phl p 2. The Phl p 2-specific IgG_1_ was used as the allergen-specific part of our bispecific antibody prototype and demonstrated blocking and protective characteristics [20,24].

Compared to the control antibody (i.e., Phl p 2 IgG_1_), which strongly inhibited allergic patients’ IgE binding to Phl p 2, the monoclonal Phl p 5 IgG_1_ did not reduce patients’ IgE binding to Phl p 5 in a relevant manner (mean inhibition of 7.2%, range: 0–12.1%), while polyclonal Phl p 5-specific rabbit serum was able to inhibit 64% (range: 41.2–93.4%) of IgE binding to Phl p 5 (Appendix A, upper part). On the contrary, the monoclonal Phl p 2-specific IgG_1_ antibody was able to reduce allergen binding of patients’ IgE antibodies by 51.9% (range: 23.5–75.7%) when compared to its control antibody (i.e., Phl p 5 IgG_1_). Polyclonal Phl p 2-specific rabbit serum blocked IgE binding by 57.8% (range: 43.3–69%) (Appendix A, lower part).

### 2.5. αPhl p 5/αICAM-1 Conjugates Protected16HBE14o- and H1HeLa Cells from RV Infection

Since human ICAM-1 is the cellular receptor for the major group RVs, we wanted to investigate if αPhl p 5/αICAM-1 conjugates protected against RV infection. For this purpose, we utilized two different well-established cell-based systems (real-time impedance-based measurements [25] and neutralization assays [19]), two different epithelial cell lines (16HBE14o- and H1HeLa [26]) and two different representatives of the major group RVs (RV-B14 and RV-A89) to monitor the influence of αPhl p 5/αICAM-1 on the epithelial barrier integrity.

For the impedance-based analysis, 16HBE14o- cells were incubated with two different concentrations of αPhl p 5/αICAM-1 conjugates (0.5 µg/mL or 1.5 µg/mL) and then exposed to 150 tissue culture infectious dose 50% (TCID)_50_/cell of RV-B14. While treatment with αPhl p 5/αICAM-1 conjugates protected cells against infection with RV-B14 in a dose-dependent manner for the complete tested period of 120 h (Figure 6, αPhl p 5/αICAM-1 + 150 TCID_50_/cell RV-B14, orange and yellow curves), incubation with medium alone could not inhibit the damage of cell monolayers (Figure 6, 150 TCID_50_/cell RV-B14, red curve). We further showed that exposure to αPhl p 5/αICAM-1 conjugates without the addition of RV-B14 did not influence cell barrier integrity at all (Figure 6, αPhl p 5/αICAM-1, light and dark blue curves).

To confirm the protective effect of αPhl p 5/αICAM-1 conjugates observed in real-time impedance-based measurements, cell-culture-based neutralization assays using H1HeLa cells, αICAM-1-IgG*Bio and RV-A89 were performed. Defined concentrations of αICAM-1-IgG*Bio were analyzed for their efficacy to inhibit damage to the cell integrity by RVs. It was revealed that already 5 µg/mL of the αICAM-1-IgG*Bio antibody fully protected cells against RV-A89 infection (Figure 7, blue bars). The stepwise reduction in the concentration of αICAM-1-IgG*Bio led to decreased protection of cells (Figure 7, blue bars). No protection against RV-A89 was seen when αPhl p 5-IgG*Strep was applied (Figure 7, green bars). Neither αICAM-1-IgG*Bio nor αPhl p 5-IgG*Strep alone had a harmful effect on H1HeLa cells (Figure 7, white bars).

## 3. Discussion

Since allergic and viral rhinitis is a global health issue, combined safe treatments of associated clinical manifestations are desirable and will eventually prevent progression to asthma exacerbations. Based on our previous findings that antibody conjugates bispecific for ICAM-1 and the major grass pollen allergen Phl p 2 were able to block allergen transmigration through the epithelial barrier [20], we aimed to explore if such bispecific antibody conjugates may also protect against RV infections through occupation of ICAM-1. Furthermore, we placed emphasis on answering the essential question concerning if any high-affinity allergen-specific antibody, independent of its ability to block IgE–allergen interaction, could be used to reduce basolateral basophil activation simply by preventing allergen uptake through the respiratory epithelium. To address both questions, we formed conjugates comprising an allergen-specific IgE-non-blocking human IgG antibody specific for Phl p 5, a highly potent major grass pollen allergen [23,27], and the same ICAM-1-specific IgG antibody used in our proof of principle study [20]. We first verified that the specificity of the αPhl p 5/αICAM-1 antibody conjugates to Phl p 5 and recombinant human ICAM-1 was retained. In addition, we demonstrated their stable and sustained binding to ICAM-1 expressed on the surface of the human respiratory epithelial cell line 16HBE14o-. The surface-bound bispecific antibody conjugates captured Phl p 5 and inhibited its uptake through the epithelial barrier. Notably, potential internalization of Phl p 5 was not found even when 16HBE14o- cells were incubated with αPhl p 5/αICAM-1+ Phl p 5 complexes for 24 h at 37 °C, confirming the results of our first model antibody conjugate (i.e., P2/ICAM1) that immobilized Phl p 2 on the cell surface over a time course of 72 h [20]. This finding is of importance because our biotin–streptavidin conjugated bispecific antibodies could contain a heterogeneous mixture of dimers and multimers that could possibly cluster ICAM-1 molecules on cell surfaces and thereby mediate endocytic uptake. Earlier studies mainly investigating endothelial cells reported that ICAM-1 undergoes more efficient endocytosis when bound by multivalent anti-ICAM-1 ligands and antibodies, whereas monovalent ligands are hardly internalized comparable to the recycling pathway of ICAM-1 in the absence of ligands [28,29,30,31,32]. However, depending on their molecular size, hydrodynamic diameter and form, not all multivalent ligands are internalized [33]. This might be one explanation for the stability of αPhl p 5/αICAM-1 antibody conjugates on the cell surface, as their hydrodynamic diameter seems to be below 100 nm [33,34,35].

Importantly, surface-bound αPhl p 5/αICAM-1 antibody conjugates significantly reduced the transmigration of free Phl p 5 through epithelial cell layers. Furthermore, no relevant amounts of Phl p 5 bound to antibody conjugates could be detected in basolateral media using a Transwell culture system. Consequently, we found significantly decreased allergen-induced basophil activation when testing basolateral media obtained from epithelial cell layers coated with αPhl p 5/αICAM-1 antibody conjugates as compared to basolateral samples from cell layers without loading αPhl p 5/αICAM-1 antibody conjugates. Our in vitro experiments were carried out with allergen doses (i.e., ng/mL) that induced strong effector cell activation after uptake of Phl p 5 through the cell layer but, nevertheless, could be captured with αPhl p 5/αICAM-1 conjugates in concentrations (i.e., µg/mL) that could be applied topically to mucous membranes of organs of allergen exposure, such as the nose or eyes.

It is remarkable that even an allergen-specific IgE-non-blocking antibody prevented allergic reactions in vitro by simply trapping the allergen before it could intrude into the mucosa. Thus, we provided strong evidence that immobilization of each clinically relevant allergen on the epithelial cell surface is, in principle, feasible with a corresponding high-affinity antibody negligible of its epitope specificity and ability to block allergen–IgE interaction. This finding is important because treatment of allergy by systemic administration of allergen-specific monoclonal antibodies is only successful when these antibodies are competing with patients’ IgE antibodies for allergen binding. Studies that investigated the efficacy of such treatments used cocktails of at least two antibodies to obtain full inhibition of allergen-induced basophil activation [36,37,38,39].

We observed an additive protective effect of our bispecific αPhl p 5/αICAM-1 antibody conjugates on RV infection of human epithelial cells in our real-time impedance-based measurements. Through cytopathogenicity assessment, we confirmed that the ICAM-1-specific antibody used for our bispecific antibody conjugate prevented epithelial barrier damage. Our results revealed that bispecific antibody conjugates were capable of inhibiting major group RV infection (exemplified by two different representatives: RV-B14 and RV-A89) by blocking human ICAM-1 receptors expressed on the surface of 16HBE14o- as well as H1HeLa cells.

Although we have to keep in mind that this effect is dependent on the epitope specificity of the ICAM-1-specific antibody, it should be possible to identify ICAM-1 antibodies, as demonstrated for 14C11 by Traub and coworkers [22] or the one we used in this study (clone 15.2, Appendix A), to achieve a synergistic protective effect in allergic patients suffering from RV-induced asthma.

As topical application has great potential for the treatment of respiratory allergies because of its painlessness and ease of administration, it should be feasible to build up a shield of protective antibodies on the apical surface of the respiratory epithelium [40]. Further effort is needed to elaborate on recombinant bispecific antibody formats, e.g., diabodies or nanobodies, to generate defined tools that continuously bind ICAM-1 to minimize applications but keep the antibody shield intact. Due to the fact that seasonal allergens are available only for short periods every year and mucosal cell surfaces are exposed to rather low allergen amounts (ng/day) [41,42,43], pollen allergy seems to be the most suitable model to develop a topical antibody-based treatment to stop allergic inflammation. Local administration of antibodies further involves the advantage of cost-effectiveness because fewer amounts of GMP-produced monoclonal antibodies are required compared to systemic administration. This fact supports the concept of topical administration as a needle-free, short-term preventive allergy and RV treatment for the future. However, further in vitro evaluation using differentiated epithelial cells and in vivo testing is required to confirm the clinical efficacy of bispecific antibody conjugates for protection against allergic and viral airway diseases.

In summary, our results demonstrated that the topical administration of bispecific antibody conjugates could be a promising approach for the treatment of airway inflammation caused by seasonal airborne allergens as well as RV infection.

## 4. Materials and Methods

### 4.1. Antigens, Rabbit Immune Sera, Human Sera, Antibodies and Antibody Conjugation

Recombinant Phl p 2 and Phl p 5 (major timothy grass pollen allergens) were obtained from Biomay AG (Vienna, Austria). Recombinant human ICAM-1 was purchased from R&D Systems (Minneapolis, MN, USA). Allergen-specific rabbit IgG antibodies were obtained by immunization of rabbits with purified recombinant allergens (Phl p 2, Phl p 5) using complete and incomplete Freund adjuvant (CFA, IFA), respectively (Charles River, Kislegg, Germany). All antibodies used for ELISA, flow cytometry and immunofluorescence staining are summarized in Appendix A. Sera were obtained from patients suffering from grass pollen allergy according to case history and IgE serology after informed consent was obtained. Serum samples were analyzed in an anonymized manner with permission from the Ethics Committee of the Medical University of Vienna (EK1641/2014). Human Phl p 2- and Phl p 5-specific IgG_1_ were expressed in Chinese hamster ovary (CHO)-K1 cells [23,24] and purified via Protein G affinity chromatography (Thermo Fisher Scientific, Waltham, MA, USA), and their concentrations were determined by measuring the UV light absorption at 280 nm based on the assumption that the absorbance value for 1 mg/mL human IgG is 1.3. The purified Phl p 5-specific IgG_1_ antibody was conjugated with Lightning-Link streptavidin (Innova Biosciences, Cambridge, UK) and termed αPhl p 5-IgG*Strep. Human ICAM-1-specific mouse IgG_1_ antibody labeled with biotin (clone 15.2) was purchased from LifeSpan BioSciences (Seattle, WA, USA) and termed αICAM-1-IgG*Bio. To form bispecific antibody conjugates, termed αPhl p 5/αICAM-1, αPhl p 5-IgG*Strep and αICAM-1-IgG*Bio (1:0.25 ratio) were co-incubated for at least 1.5 h (h) at room temperature and overnight (o.n.) at 4 °C.

### 4.2. ELISA Evaluation of the Binding Specificities of αPhl p 5/αICAM-1 Antibody Conjugates

ELISA plates (Nunc MaxiSorp, Roskilde, Denmark) were coated with recombinant human ICAM-1 (R&D Systems) or recombinant Phl p 5 (Biomay AG) (2 µg/mL in 100 mM NaHCO_3_, pH 9.6). Plates were incubated for 1 h at 37 °C, washed twice with 1x phosphate-buffered saline (1x PBS) containing 0.05% (*v/v*) Tween-20 (PBST) and blocked with PBST containing 3% (*w/v*) bovine serum albumin (BSA) for 3 h. Increasing concentrations of αPhl p 5/αICAM-1 conjugates (0.2 µg/mL, 1 μg/mL and 5 μg/mL) and, for control purposes, αPhl p 5-specific IgG*Strep (1 µg/mL) and αICAM-1-specific IgG*Bio (1 µg/mL) were applied o.n. at 4 °C. Bound antibodies were detected either with alkaline phosphatase (AP)-conjugated rat anti-mouse IgG_1_ antibodies (BD Pharmingen, San Jose, CA, USA) diluted 1:5000 or with AP-conjugated goat anti-human IgG F(ab’)_2_ antibodies (Thermo Fisher Scientific) diluted 1:1000 in PBST/0.5% (*w/v*) BSA. The color reaction was obtained with phosphatase substrate (1 mg/mL, Sigma-Aldrich, St. Louis, MO, USA). OD measurements were carried out on ELISA reader Infinite F50 (Tecan, Männedorf, Switzerland) at 405 nm (reference wavelength: 550 nm) with an integrated software, i-control 2.0. Results are shown as mean values of duplicates of five independent experiments. Error bars indicate SD.

### 4.3. Cultivation of the Human Epithelial Cell Lines 16HBE14o- and H1HeLa

The epithelial cell line 16HBE14o-, derived from human bronchial epithelial cells (Prof. D.C. Gruenert, University of California, San Francisco, CA, USA), was used as a surrogate for the respiratory epithelium. The cells show properties of differentiated airway epithelial cells including formation of tight junctions, apical microvilli and cilia and form polarized monolayers when grown on Transwell filters [44]. The 16HBE14o- cells were cultured at 37 °C in a humidified atmosphere containing 5% CO_2_ in Minimum Essential Medium (MEM; Gibco, Thermo Fisher Scientific) supplemented with 10% fetal bovine serum (FBS; HyClone, GE Healthcare, Buckinghamshire, UK), 100 U/mL penicillin and 100 µg/mL streptomycin (Gibco, Thermo Fisher Scientific). Cells were cultured in flasks coated with LHC basal medium (Gibco, Thermo Fisher Scientific) supplemented with 100 µg/mL BSA, 30 µg/mL collagen and 10 µg/mL fibronectin (BD Biosciences, San Jose, CA, USA).

The cervical epithelial cell line H1HeLa (American Type Culture Collection, ATCC, Manassas, VA, USA) was grown at 37 °C in a humidified atmosphere containing 5% CO_2_ in MEM supplemented with 10% FBS (Gibco, Thermo Fisher Scientific) and antibiotics (100 U/mL penicillin, 100 µg/mL streptomycin and 1.2 µg/mL gentamycin (Gibco, Thermo Fisher Scientific).

### 4.4. Flow Cytometry

16HBE14o- cells (2 × 10^5^ per well) seeded in 96-well V-bottom plates (Greiner, Kremsmünster, Austria) were washed with 1x PBS/0.5% (*w/v*) BSA and stained with Fixable Viability Dye eFlour 780 (Thermo Fisher Scientific) for 20 min to exclude dead cells from analysis. After washing cells again, they were blocked with 10% goat serum (Thermo Fisher Scientific) in 1x PBS/0.5% (*w/v*) BSA for 20 min. After blocking, cells were washed and incubated with 1 μg αICAM-1 IgG*Bio (LifeSpan BioSciences) or 1 µg anti-human IgA_1_/IgA_2_*Biotin (BD Bioscience) as the isotype control in 50 µL 1x PBS/0.1% (*w/v*) BSA. After incubation for 20 min on ice and in darkness, cells were stained with streptavidin DyLight 488 (Thermo Fisher Scientific) and analyzed on a Canto II Cytometer (BD Bioscience) counting at least 50,000 cells per sample in triplicates. Results were evaluated with FlowJo Software (version 10.7.2, (Tree Star Inc., Ashland, OR, USA).

In a second set of experiments, 16HBE14o- cells (2 × 10^5^ per well) were blocked and stained for alive/dead distinction as described above. Then, cells were washed and incubated with 1 µg αPhl p 5/αICAM-1 in 50 μL 1x PBS/0.5% (*w/v*) BSA for 30 min on ice. After washing, aliquots of 1 µg, 2 µg or 5 μg Phl p 5 in 50 μL 1x PBS supplemented with 0.1% (*w/v*) BSA were added, and cells were again incubated for 30 min on ice. After further washing, cells were incubated with Phl p 5-specific rabbit antibodies or pre-immune rabbit serum as the isotype control (Charles River) diluted 1:1000 in 1x PBS/0.1% (*w/v*) BSA. Staining was performed with Alexa Fluor 405-labeled goat anti-rabbit antibody (10 µg/mL; Thermo Fisher Scientific). Cells were analyzed on a Canto II Cytometer (BD Bioscience) counting at least 50,000 cells per sample in triplicates. Results were evaluated with FlowJo Software (version 10.7.2).

### 4.5. Immunofluorescence Microscopy

16HBE14o- cells (2 × 10^4^ cells per well) were seeded on ibiTreat tissue culture-treated 8-well μ-slides (Ibidi GmbH, Munich, Germany) and grown o.n. to approximately 90% confluency. Complexes of αPhl p 5/αICAM-1 + Phl p 5 were formed by co-incubation of 5 µg αPhl p 5/αICAM-1 and 1 µg Phl p 5 in 300 µL MEM (supplemented with 10% FBS, 100 U/mL penicillin and 100 µg/mL streptomycin) for 1 h at room temperature. Cells were washed three times with 1x PBS and afterwards incubated with αPhl p 5/αICAM-1 + Phl p 5 complexes at either 4 °C for 1 h or 37 °C for 24 h. For control purposes, cells were incubated with 5 µg αPhl p 5/αICAM-1 or 1 µg Phl p 5 only. Unbound reagents were removed by washing cells three times with 1x PBS. Cells were fixed with 4% formaldehyde solution (Invitrogen, Carlsbad, CA, USA) in 1x PBS for 20 min at room temperature and washed with 1x PBS. Remaining aldehyde groups were quenched with 50 mM ammonium chloride in 1x PBS for 10 min. Unspecific binding sites were blocked with 10% (*v/v*) goat serum (Jackson ImmunoResearch Laboratories, West Grove, PA, USA) in 1x PBS overnight at 4 °C. To permeabilize cell membranes, thus enabling detection of intracellular αPhl p 5/αICAM-1 or Phl p 5, 0.05% saponin was added to the blocking buffer. After washing, cells were then incubated with Phl p 5-specific rabbit antibodies (Charles River) diluted 1:1000 in 1x PBS/0.05% (*w/v*) saponin for 1 h at room temperature. To detect αPhl p 5/αICAM-1, Alexa Fluor 488 goat anti-mouse IgG (Molecular Probes, Eugene, OR, USA) diluted 1:2000 in blocking buffer was added to the cells for 2 h at room temperature. Phl p 5 bound to αPhl p 5/αICAM-1 detected by Phl p 5-specific rabbit antibodies was visualized with Alexa Fluor 568 goat anti-rabbit IgG (Molecular Probes) applied at a dilution of 1:2000 in blocking buffer for 2 h at room temperature. Finally, nuclei were stained either with 1 µg/mL DAPI (Molecular Probes) or with 25 µM 1,5-bis{[2-(di-methylamino) ethyl]amino}-4, 8-dihydroxyanthracene-9,10-dione (DRAQ5) (Thermo Fisher Scientific) in 1x PBS for 20 min at room temperature. Cells were stored in 1x PBS at 4 °C until imaging. Wide-field fluorescence imaging of fixed cells was carried out using a Zeiss Observer Z1 Axio inverted fluorescence microscope equipped with an oil immersion 40x objective (Pan Apochromat, 1.4 NA, Carl Zeiss Inc., Oberkochen, Germany), the Zeiss Axio Software package (version 4.9.1) and appropriate filter sets to excite and detect the fluorescence emission of DAPI, Alexa Fluor 488 and Alexa Fluor 568. Confocal images were acquired using an UltraVIEW ERS Confocal Imager (Perkin Elmer, Waltham, MA, USA) connected to a Zeiss Axiovert 200 microscope fitted with a 63x/1.4 oil objective lens (Plan-Apochromat, Zeiss). Alexa Fluor 488 and 568 fluorophores as well as DRAQ5 were excited at 488, 568 or 647 nm, respectively, using a 488/568/647 multiline argon/krypton laser. Pictures were digitized and processed with Volocity software (Version 5.5., Perkin Elmer). Individual representative images were further processed with Adobe Photoshop using identical conditions for positive and negative controls.

### 4.6. Transwell Assay

16HBE14o- cells were cultured either in 6.5 mm (1 × 10^5^ cells per support) or in 12 mm (2 × 10^5^ cells per support) collagen–fibronectin-coated permeable supports (polyester membranes, 0.4 µm pore size, Costar, Corning, NY, USA) preloaded with 0.1 mL (for 6.5 mm inserts) or 0.5 mL (for 12 mm inserts) MEM supplemented with FBS, penicillin and streptomycin, as described above. The medium in supports (apical chamber) is further referred to as “apical medium”. The lower (basolateral) chamber was filled with 0.6 mL or 1.5 mL of medium (further referred to as “basolateral medium”). The trans-epithelial electrical resistance (TEER) baseline was measured with an ohm voltmeter (Millipore, Bedford, MA, USA or World precision instruments, Inc. Sarasota, FL, USA) and ranged consistently from 120 to 140 Ωcm^2^ for inserts filled with medium but without cells. When cells reached a TEER value of at least 500 Ωcm^2^ (corresponding to polarized monolayers) [45], 50 ng/mL interferon-γ (Pepro Tech Inc, Rocky Hill, NJ, US) was added to the basolateral medium to mimic in situ conditions [45] and allow for detectable allergen transmigration through the cell monolayer [20,45]. When TEER values were below 200 Ωcm^2^, 10 µg/mL (2 independent experiments performed in duplicates) or 20 µg/mL (3 independent experiments performed in duplicates) αPhl p 5/αICAM-1 was added to the inserts for 3 h at 37 °C for binding to ICAM-1 expressed on the cell surface. For control purposes, cells were incubated in parallel for 3 h at 37 °C with medium only. Thereafter, apical media of all wells were removed and stored at −20 °C for later analysis. Cells were washed once with medium to remove unbound αPhl p 5/αICAM-1. Then, 10 ng/mL (2 independent experiments performed in duplicates) or 20 ng/mL (3 independent experiments performed in duplicates) Phl p 5, or 10 ng/mL Phl p 2 (1 experiment performed in duplicates) diluted in medium was added to the inserts, and cells were incubated for 24 h at 37 °C. Finally, apical and basolateral media were collected and analyzed. The effect of αPhl p 5/αICAM-1 on the apical-to-basolateral trans-epithelial migration of Phl p 5 and Phl p 2 was measured by allergen quantities in basolateral media and corresponding apical media. Detected allergen amounts in basolateral media deriving from cells coated with or without αPhl p 5/αICAM-1 were compared. In addition to the determination of free Phl p 5 or Phl p 2 (4.7), the penetration of Phl p 5 bound to αPhl p 5/αICAM-1 through the membranes was assessed by ELISA experiments (4.8).

### 4.7. ELISA Measuring Free Phl p 5 in Apical and Basolateral Media

To measure free Phl p 5 or Phl p 2 in apical and basolateral media, ELISA plates (Nunc MaxiSorp) were coated with Phl p 5-specific IgG_1_ [23] or Phl p 2-specific IgG_1_ [24] (1 µg/mL in 100 mM NaHCO_3_, pH 9.6, 1 h, 37 °C or o.n. 4 °C). Plates were then washed twice with PBST and saturated with PBST containing 3% (*w/v*) BSA. Aliquots of apical and basolateral media deriving from 16HBE14o- cell monolayers treated +/− αPhl p 5/αICAM-1 (diluted 1:24 and 1:4 in PBST/0.5% (*w/v*) BSA, respectively) were applied o.n. at 4 °C. Phl p 5 or Phl p 2 was then detected with Phl p 5- or Phl p 2-specific rabbit antibodies (1:1000 in PBST/0.5% (*w/v*) BSA). Bound rabbit antibodies were visualized with a horseradish peroxidase (HRP)-conjugated donkey anti-rabbit antibody diluted 1:2000 in PBST/0.5% (*w/v*) BSA (Cytiva). OD measurements at 405 nm with a reference wavelength of 490 nm were performed as described above (4.2). Results are shown as mean values of triplicates, and error bars indicate SD.

### 4.8. ELISA Measuring Phl p 5 Bound to αPhl p 5/αICAM-1 in Apical and Basolateral Media

To detect αPhl p 5/αICAM-1-bound Phl p 5, ELISA plates (Nunc MaxiSorp) were coated with recombinant human ICAM-1 (2.5 µg/mL in 100 mM NaHCO_3_, pH 9.6, o.n., 4 °C) that were washed and blocked as described above. Aliquots of apical and basolateral media deriving from 16HBE14o- cell monolayers treated +/− αPhl p 5/αICAM-1 (diluted 1:24 and 1:4 in PBST/0.5% (*w/v*) BSA, respectively) were added, and αPhl p 5/αICAM-1-bound Phl p 5 was detected with rabbit anti-Phl p 5 antibodies as described above. To test the functionality of our developed ELISA assay, preformed αPhl p 5/αICAM-1 + Phl p 5 complexes as applied for immunofluorescence microscopy were diluted 1:5 in PBST/0.5% (*w/v*) BSA and added to ICAM-1-coated plates. OD measurements at 405 nm with a reference wavelength of 490 nm were performed as described above. Results are shown as mean values of triplicates, and error bars indicate SD.

### 4.9. Rat Basophilic Leukemia Cell-Based Mediator-Release Assay

Humanized rat basophilic leukemia (RBL) cells transfected with the human high-affinity IgE receptor FcεRI (RS-ATL8) [46] were cultured in MEM (Thermo Fisher Scientific) containing 10% FBS, 2 mM L-glutamine, 100 U/mL penicillin, 100 µg/mL streptomycin, 0.2 µg/mL hygromycin B and 0.2 µg/mL geneticin (all supplements from Gibco) at 37 °C in a humidified atmosphere containing 5% CO_2_. Cells at 80% confluency were transferred from tissue culture flasks into transparent, flat-bottomed, sterile 96-well cell culture plates (Costar) (1.5 × 10^5^ cells per well). Cells were incubated with specific IgEs from the serum of grass pollen-allergic patients diluted 1:10 o.n. at 37 °C. After washing three times with 2x Tyrode’s buffer (Sigma-Aldrich), cells were exposed to different dilutions of basolateral media (1:10–1:40) for 1 h at 37 °C obtained from 16HBE14o- cell monolayers treated +/− αPhl p 5/αICAM-1. RS-ATL8 cells were incubated with 2x Tyrode’s buffer only to measure spontaneous release. Controls omitting either patients’ sera or basolateral media were also included. Total β-hexosaminidase content was determined after lysis of the cells by the addition of Triton-X100 (Sigma-Aldrich, final concentration: 1%). After adding the substrate 4-Methylumbelliferyl N-acetyl-β-D-galactosaminide (4-MUG), β-hexosaminidase amounts were measured by fluorescence (excitation wavelength: 360 nm; emission wavelength: 465 nm) with a TECAN infinite M200pro plate reader and are reported as the percentage of total mediator content.

### 4.10. ELISA Determining the Capacity of Phl p 2- and Phl p 5-Specific IgG_1_ Antibodies to Inhibit Patients’ IgE Binding

ELISA plates (Nunc, Maxisorp) were coated with recombinant Phl p 5 and Phl p 2 (Biomay) (0.2 µg/mL in 100 mM NaHCO_3_, pH 9.6) o.n. at 4 °C. The next day after washing, plates were blocked with PBST/3% (*w/v*) BSA at 37 °C for 3 h and incubated with 20x excess (20 µg/mL for Phl p 5-coated wells and 50 µg/mL for Phl p 2-coated wells) of Phl p 2 IgG_1_ [24] and Phl p 5 IgG_1_ [23] o.n. at 4 °C. For control purposes, Phl p 2- and Phl p 5-specific rabbit sera diluted 1:100 in PBST/0.5% (*w/v*) BSA were applied. After washing, sera from grass pollen-allergic patients diluted 1:10 (patient 8) or 1:5 (patients 9–12) were added o.n. Bound patients’ IgE antibodies were detected with AP-conjugated mouse anti-human IgE antiserum (BD Pharmingen) diluted 1:1000 in PBST/0.5% BSA (*w/v*). ODs were measured at 405 nm with a reference wavelength of 550 nm as described above. All the determinations were performed in triplicates and are shown as mean values.

### 4.11. Preparation of RV Strains

RV-B14 and RV-A89 (ATCC) were amplified in H1HeLa cells and purified as previously described [47]. Purified RV-B14 and RV-A89 were then used for the infection of confluent epithelial cell monolayers in e-plates of the xCELLigence DP system or in 96-well tissue culture plates (Corning) for the neutralization assay, respectively. RV-B14 and RV-A89 titers were determined by a 50% tissue culture infectious dose (TCID_50_) assay [48].

### 4.12. xCELLigence Real-Time Cell Analysis DP System

16HBE14o- cells were transferred at 70–90% confluency from tissue culture flasks into collagen–fibronectin-coated wells of e-plates of the xCELLigence Real-Time Cell Analysis DP system (ACEA Biosciences, San Diego, CA, USA). An amount of 200 µL of cell suspensions per well was cultured at a concentration of 1 × 10^5^ cells/mL. Real-time cell electronic sensing (RT-CES) is a label-free technique for automatic and continuous electronic monitoring of adherent living cells. This measuring method monitors non-invasive impedance-based cell responses at physiological conditions. The outcome is complementary to, e.g., cell proliferation, morphology changes and cytotoxicity. The cell index is expressed as an arbitrary unit and is calculated from impedance measurements between cells and sensors of the xCELLigence DP system. The confluency of cell monolayers was confirmed by phase-contrast microscopy (IX73 Olympus, Tokyo, Japan) and is equivalent to cell index values of 13–15, i.e., approximately after 20 h of incubation. At this time point, the cell index values were normalized (normalized cell index = 1), i.e., 30 min before the addition of the respective compounds (αPhl p 5/αICAM-1 conjugates or medium) to the epithelial cell monolayers in MEM containing 1% (*v/v*) FBS. Three hours later, the cells under various conditions were infected with 150 TCID_50_/cell concentrations of RV-B14 or mock-infected by adding PBS. Changes in cell responses were measured and recorded every 30 min [25].

### 4.13. Cell-Culture-Based RV Neutralization Assay

H1HeLa cells were seeded in 96-well cell culture plates at a density of 1.3 × 10^4^ cells/well and grown o.n. to 70–80% confluency as described [47,49]. αICAM-1 IgG*Bio and αPhl p 5 IgG*Strep (as control) were prepared in 2-fold serial dilutions (between 5 µg/mL and 0.03125 µg/mL) in MEM containing 1% (*v/v*) FBS and 30 mM MgCl_2_. An amount of 100 µL of each dilution was added to H1HeLa cells in triplicates and incubated for 3 h at 37 °C to enable the binding of specific antibodies to ICAM-1 on the surface of the cells. After incubation, monoclonal antibodies were removed and 100 µL of RV-A89 (100 TCID_50_) was added to the plates and incubated at 34 °C for three days. Afterwards, the virus was removed and cells were stained with 0.1% (*w/v*) crystal violet for 10 min. For control purposes, cells were incubated with RV-A89, medium, αICAM-1 IgG*Bio or αPhl p 5 IgG* Strep only. For quantification, the crystal violet stain was subsequently dissolved in 30 µL of 1% (*v/v*) SDS solution (ddH_2_O), and the plates were read out at OD 560 nm on a TECAN Infinite F50 ELISA reader with an integrated software, i-control 2.0 (Tecan) [47].

### 4.14. Statistical Analysis

Differences in the amounts of free Phl p 5 (ELISA experiments) or in the percentage of Phl p 5-induced ß-hexosaminidase release of RBL cells (RBL mediator-release assays) among the two groups (with or without treatment of cells with αPhl p 5/αICAM-1 conjugates) were analyzed with two-tailed Mann–Whitney U tests. Results with a *p*-value < 0.05 were considered significant (* *p* < 0.05 and ** *p* < 0.01). Statistical analyses were performed with GraphPad Prism Version 5.0 (GraphPad Software Inc., San Diego, CA, USA).

## Figures and Tables

**Figure 1 ijms-24-02725-f001:**
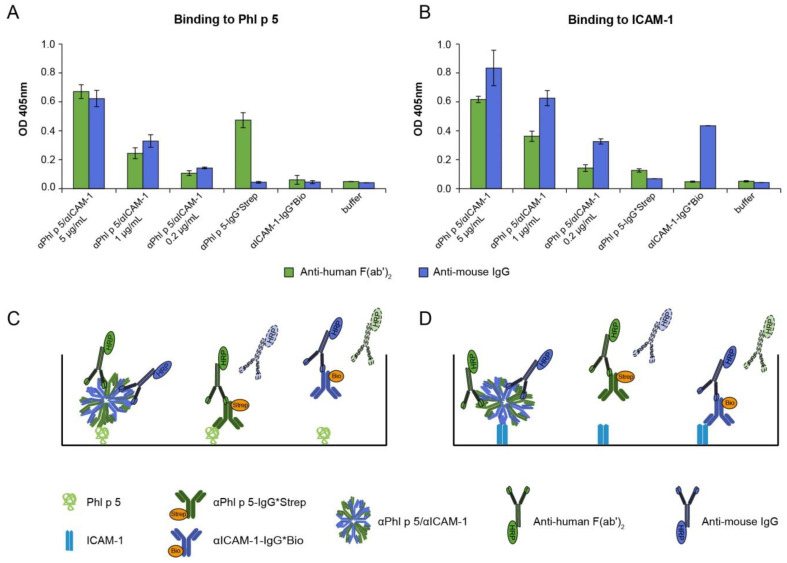
Antibody conjugates (αPhl p 5/αICAM-1) bind to Phl p 5 and human ICAM-1. Conjugates formed with αPhl p 5-IgG*Strep and αICAM-1-IgG*Bio (*x*-axes: αPhl p 5/αICAM-1; 5 µg/mL, 1 µg/mL and 0.2 µg/mL), αPhl p 5-IgG*Strep or αICAM-1-IgG*Bio alone were tested for reactivity to Phl p 5 (**A**) and ICAM-1 (**B**) and detected by anti-human F(ab´)_2_ (green bars) or anti-mouse IgG (blue bars). Optical density (OD) values (*y*-axes) corresponding to bound αPhl p 5/αICAM-1 conjugates, αPhl p 5-IgG*Strep or αICAM-1-IgG*Bio are shown as means of 10-fold analysis (deriving from duplicate determination of five independent experiments) ± standard deviation (SD)s. (**C**,**D**) are schematic representations of the experiments in (**A**,**B**).

**Figure 2 ijms-24-02725-f002:**
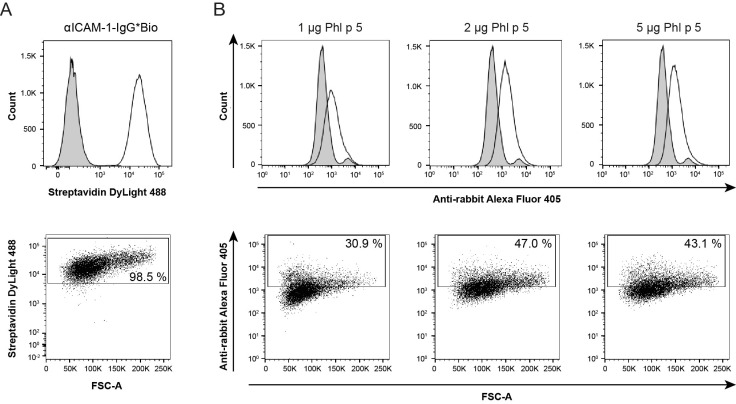
Detection of ICAM-1 or Phl p 5 captured by αPhl p 5/αICAM-1 conjugates on 16HBE14o- cells through flow cytometric analysis. (**A**) Expression of human ICAM-1 on the surface of 16HBE14o- cells was confirmed by using αICAM-1-IgG*Bio (**upper** panel: white) or an isotype control (**upper** panel: gray) and is shown as the percentage of ICAM-1-positive cells out of the alive cells in the scatter plot (**lower** panel). (**B**) Cells were incubated with αPhl p 5/αICAM-1 conjugates and different amounts of Phl p 5 (1 µg/50 µL, 2 µg/50 µL and 5 µg/50 µL) and were subsequently probed for αPhl p 5/αICAM-1-bound Phl p 5 with specific rabbit serum or pre-immune serum as an isotype control. Results are displayed as overlayed histograms of Phl p 5-specific rabbit serum (white) and the isotype control (gray) (**upper** panels), or as percentages of Phl p 5-positive cells out of the alive cells in the scatter plots (**lower** panels). Scatter plots depict the forward scatter (FSC-A) (*x*-axes) against (*y*-axes) (**A**) Steptavidin Dylight 488 and (**B**) anti-rabbit Alexa Fluor 405. Experiments were performed in triplicates, and displayed data are representatives of two independent experiments.

**Figure 3 ijms-24-02725-f003:**
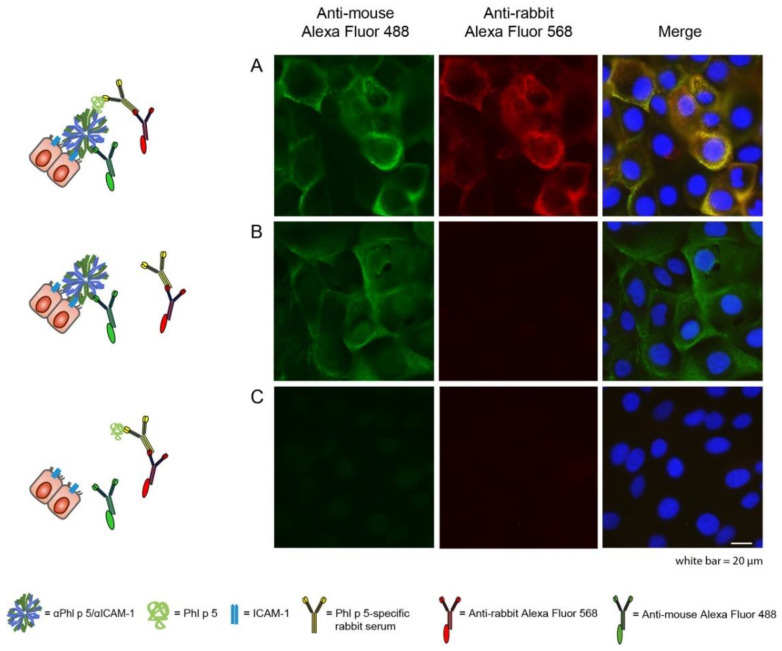
Visualization of αPhl p 5/αICAM-1 conjugates and Phl p 5 on 16HBE14o- cells by immunofluorescence microscopy. Cells were incubated with αPhl p 5/αICAM-1 (**A**,**B**) and/or Phl p 5 (**A**,**C**) and stained with Alexa Fluor 488-labeled anti-mouse antibodies (**left** column, green) and Alexa Flour 568-labeled anti-rabbit antibodies (**middle** column, red) to visualize αPhl p 5/αICAM-1 and Phl p 5, respectively. Nuclei were stained with 4′,6-diamidino-2-phenylindole dihydrochloride (DAPI) (blue), and merged images are shown in the **right** column. Scale bar is 20 µm. Experiments were performed in duplicates. Data shown are representatives of two independent experiments.

**Figure 4 ijms-24-02725-f004:**
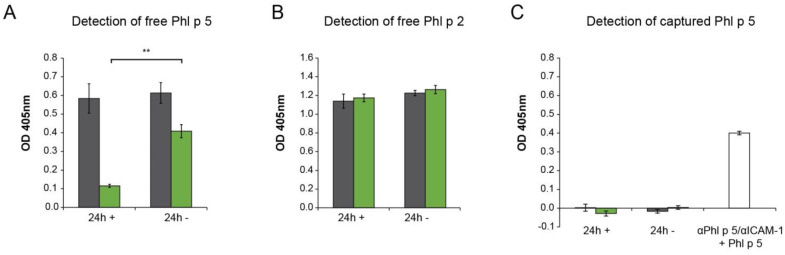
Epithelial-cell-bound αPhl p 5/αICAM-1 conjugates significantly reduce penetration of Phl p 5 but not of Phl p 2, an unrelated grass pollen allergen. Detection of free (**A**) Phl p 5 and (**B**) Phl p 2 or conjugate-captured (**C**) Phl p 5 in apical (gray bars) and basolateral media (green bars) of 16HBE14o- cell monolayers that had been loaded with (**+**) or without (**−**) αPhl p 5/αICAM-1 conjugates. OD values corresponding to concentrations of free (**A**) Phl p 5 and (**B**) Phl p 2, or bound (**C**) Phl p 5 measured 24 h after +/− application of conjugates onto 16HBE14o- cells are shown as means of triplicates ± SDs. Depicted values are background-subtracted. The value ** *p* < 0.01 shows significant difference. To verify the assay functionality performed for conjugate-bound Phl p 5 (**C**), preformed complexes of αPhl p 5/αICAM-1 and Phl p 5 were applied. Representative bars of five independent experiments (**A**,**C**) are shown.

**Figure 5 ijms-24-02725-f005:**
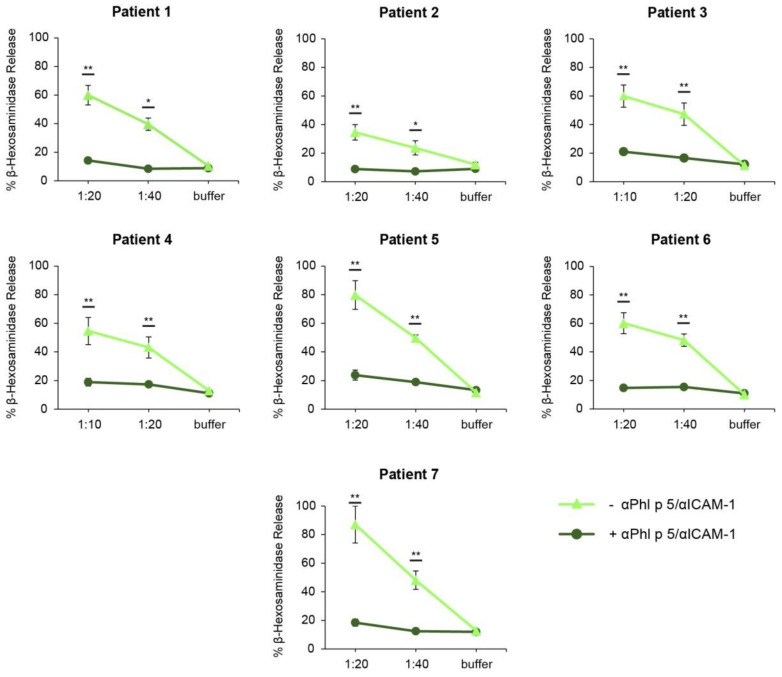
Inhibition of trans-epithelial migration of Phl p 5 results in decreased basophil activation. Rat basophilic leukemia (RBL) cells expressing the human FcεRI were pre-incubated with the sera of 7 grass pollen-allergic patients. Consequently, buffer (buffer, *x*-axes) or basolateral media deriving from cell monolayers treated with (**+**) or without (**−**) αPhl p 5/αICAM-1 conjugates were applied at two different dilutions (1:10–1:40; *x*-axes). The percentage of β-hexosaminidase release induced by Phl p 5 is displayed on the *y*-axes in relation to the total ß-hexosaminidase amount of lysed cells. Displayed graphs are representative of RBL cell assays of five independent Transwell experiments performed in duplicates. Values are shown as means of 6-fold analysis (deriving from triplicate determination of duplicates) ± SDs. The values * *p* < 0.05 and ** *p* < 0.01 show significant difference.

**Figure 6 ijms-24-02725-f006:**
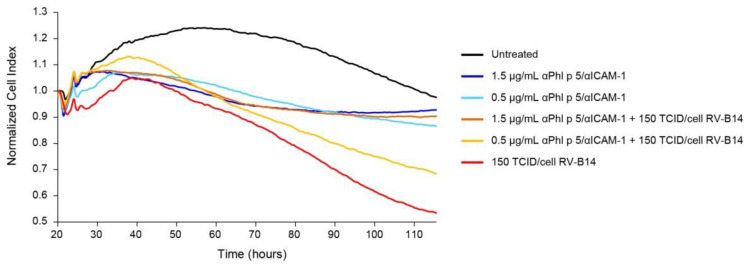
Treatment with αPhl p 5/αICAM-1 conjugates protects epithelial cells against damage by RV infection. The 16HBE14o- cells were cultured on e-plates of the xCELLigence DP system and were incubated with different concentrations of αPhl p 5/αICAM-1 conjugates (0.5 µg/mL and 1.5 µg/mL) or medium. Three hours thereafter, cells were infected with 150 TCID_50_/cell of RV-B14. Cells that were not infected with RV-B14 served as controls (untreated, 0.5 µg/mL and 1.5 µg/mL αPhl p 5/αICAM-1). Impedance values were automatically measured every 30 min (min) by the xCELLigence DP system for 120 h and are expressed as a normalized cell index (*y*-axis). Data are shown as mean values of triplicates for each measuring point and are representative of two independent experiments.

**Figure 7 ijms-24-02725-f007:**
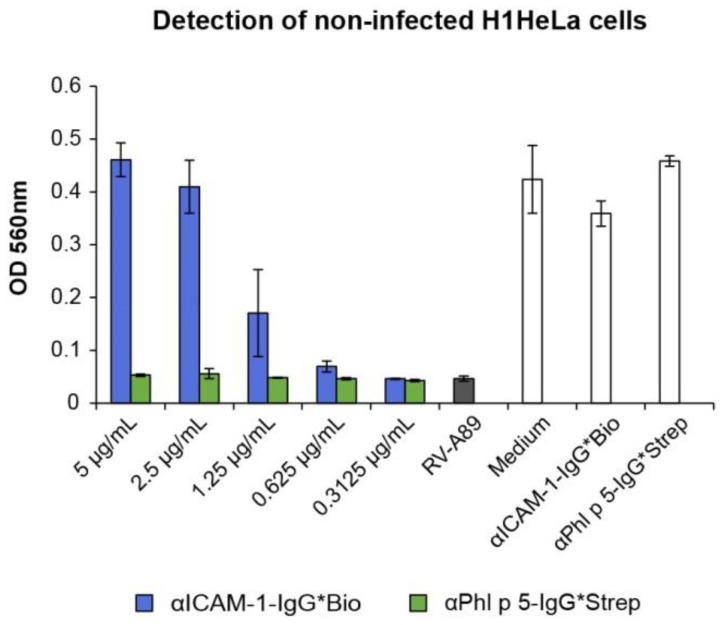
Inhibition of RV infection by αICAM-1-IgG*Bio determined by cell-culture-based neutralization assays. H1HeLa cells were pre-incubated with 2-fold serial dilutions of αICAM-1-IgG*Bio (blue bars) and, for control purposes, with αPhl p5-IgG*Strep (green bars) or medium. After 3 h, 100 TCID_50_/cell of RV-A89 was added, and cells were incubated for 3 days. Cells incubated with virus only (gray bar; RV-A89), medium only (white bar; medium) or αICAM-1-IgG*Bio and αPhl p5-IgG*Strep without virus (white bars) served as controls. Remaining cells were stained with crystal violet and OD was read at 560 nm (*y*-axis). Data are shown as the mean values of triplicates and are representative of three independent experiments.

## Data Availability

Not applicable.

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
