# Peer review of "Antibody Conjugates Bispecific for Pollen Allergens and ICAM-1 with Potential to Prevent Epithelial Allergen Transmigration and Rhinovirus Infection"

_ijms, 2023, doi:10.3390/ijms24032725_

Round 1

Reviewer 1 Report

The authors provide evidence that bivalent antibody conjugates are more efficient in preventing activation of basophils by allergens. They also show that RV entry can be prevented by these bivalent antibodies.  The presented results in the manuscripts are technically sound and logically presented.

Major comments:  16HBE14oC although polarize and make tight junctions, they do not differentiate and therefore the results cannot be translated to in vivo situation.  Using mucociliary-dffierentiated cells to confirm the results is strongly recommended. 

Patients' sera are used in this manuscript.  Please provide ethical statement and also details on the IRB approval

What is the rationale for using RV14 in one experiment and RV89 in others.  

Reviewer 2 Report

Comments

In this manuscript, the authors have synthesized antibody conjugates bispecific for ICAM-1 and Phl p 5 and analyzed their capacity to affect allergen penetration and RV infection. There results indicate the potential of allergen/ICAM-1-specific antibody conjugates as a topical treatment strategy for allergy and RV infections. The research topic is interesting, the whole manuscript is well structured. The authors have performed several in vitro studies to support their hypothesis. While this work falls within the scope of IJMS, the authors must address the following issues before further consideration: -

1.     Although there are more experiments added, this research work is somewhat similar with previously published work, except for Phl p 2 replacing with Phl p 5: https://www.ncbi.nlm.nih.gov/pmc/articles/PMC4530582/

2.     In the figure 1, the authors mentioned that shown data are representatives of five independent experiments. However, there is no statistical descriptions. Please include statistical analysis, standard deviation, or error bar in the Figure 1a and 1b.

3.     On page 4, lines 144-148, The authors mentioned that around 31% of the cells had Phl p 5 bound to their cell surface and reached its peak (47%) when the amount of Phl p 5 was increased to 2 μg, but it could not be further enhanced when 5 μg Phl p 5 was applied. The authors must be discussed in detail why this phenomenon happened, is it due to saturation of active site/binding site or any other reasons?

Round 2

Reviewer 2 Report

The authors have included all my concerns in the revised manuscript and I recommend accepting this manuscript for publication in its current form.